# PEL-NAS: Search Space Partitioned Architecture Prompt Co-Evolutionary LLM-driven Hardware-Aware Neural Architecture Search

## Abstract

Hardware-Aware Neural Architecture Search (HW-NAS) requires joint optimization of accuracy and latency under device constraints. Traditional supernet-based methods require multiple GPU days per dataset. Large Language Model (LLM)-driven approaches avoid training a large supernet and can provide quick feedback, but we observe an *exploration bias*: the LLM repeatedly proposes neural network designs within limited search space and fails to discover architectures across different latency ranges in the whole search space. To address this issue, we propose **PEL-NAS**: a search space **P**artitioned, architecture prompt co-**E**volutionary and **L**LM-driven **N**eural **A**rchitecture **S**earch that can generate neural networks with high accuracy and low latency with reduced search cost. Our proposed PEL-NAS has three key components: 1) a complexity-driven partitioning engine that divides the search space by complexity to enforce diversity and mitigate exploration bias; 2) an LLM-powered architecture prompt co-evolution operator, in which the LLM first updates a knowledge base of design heuristics based on results from the previous round, then performs a guided evolution algorithm on architectures with prompts that incorporate this knowledge base. Prompts and designs improve together across rounds which avoid random guesswork and improve efficiency; 3) a zero-cost predictor to avoid training a large number of candidates from scratch. Experimental results show that on HW-NAS-Bench, PEL-NAS can achieve overall higher HV, lower IGD, and up to **54%** lower latency than baselines at similar accuracy. Meanwhile, the search cost drops from days to minutes compared with traditional supernet baselines.

## 1 Introduction

As deep learning expands into resource-constrained environments such as the Internet of Things (IoT) devices, Hardware-Aware Neural Architecture Search (HW-NAS) becomes essential for discovering models that optimize the trade-off between accuracy and inference latency Benmeziane et al. (2021b;a). Supernet-based paradigm, such as Once-for-All (OFA) Cai et al. (2019) and Fair-NAS Chu et al. (2021), achieve strong performance but require extensive computational resources. For example, FairNAS requires about 10 GPU-days to train a supernet on a V100 for ImageNet Benmeziane et al. (2023). This has driven interest in training-free NAS methods, such as SynFlow Tanaka et al. (2020), Fisher Theis et al. (2018), and Jacobian Covariance Mellor et al. (2021), which can rank untrained networks using zero-cost proxies, without requiring full training.

Recently, Large Language Models (LLMs) offer a promising training-free alternative for discovering neural architectures Achiam et al. (2023). However, applying an LLM directly to the vast HW-NAS search space raises two challenges. First, we observe the exploration bias issue, which is analogous to the mode collapse issue in generative models Shumailov et al. (2024); Kossale et al. (2022); Zhang et al. (2025). Specifically, the LLM tends to repeatedly generate safe and familiar architectural patterns within limited search space, without fully exploring the full search space. Figure 1 compares three generation strategies on HW-NAS-Bench (Edge GPU, CIFAR-10). In *(a) Normal prompt*, we give only a plain task description including target device and dataset and ask the

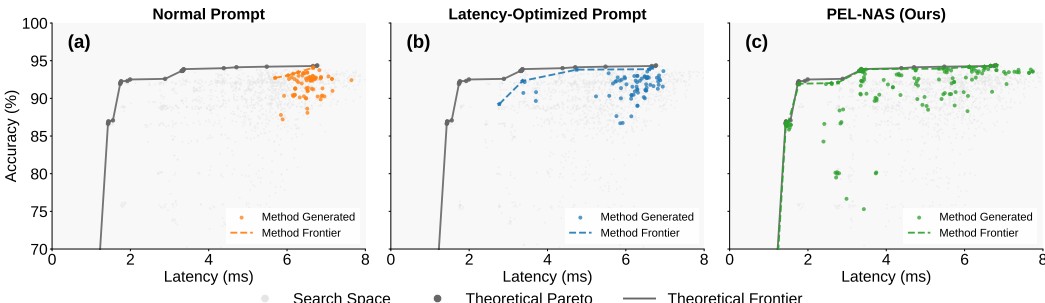

Figure 1: Comparison of three generation strategies on HW-NAS-Bench (Edge GPU, CIFAR-10): normal prompt (orange), latency-optimized prompt (blue), and PEL-NAS (green). Latency-optimized prompting increases coverage compared to standard prompting but still leaves gaps, while PEL-NAS achieves near-complete coverage across latency ranges.

LLM to propose an architecture. The LLM then concentrates in a small area with limited coverage of the latency range. In *(b) Latency-optimized prompt*, we add an explicit hint to aim for diverse latencies and pass back the previous round's accuracy and latency to the LLM. The results shift toward lower latency but the coverage remains uneven. The number of low-latency architectures attempted by LLM is still small and not competitive. This motivates the development of a strategy that can further expand search space. Second, most existing LLM approaches rely on static prompts, lacking a mechanism to accumulate knowledge from past evaluations. Without this feedback loop, the LLM cannot refine its design rules over generations, which slows progress toward the true Pareto front.

To address the above two challenges, we propose **PEL-NAS**: a search space **P**artitioned, architecture prompt co-**E**volutionary and **L**LM-driven **N**eural **A**rchitecture **S**earch (Figure 2), to reduce exploration bias while improving search efficiency. Our approach begins with a a complexity-driven partitioning strategy that decomposes the vast search space into subspaces with different complexity or different parameter size levels. With the partitioning strategy, PEL-NAS can discover subnetworks across the whole search space, as shown in Figure 1(c). Within each subspace, we then employ an LLM-Powered Evolutionary Operator that functions as an expert reasoning engine, guided by a continually refined Co-evolve Knowledge Base. For each new design, the LLM provides a detailed rationale for its modifications, and a rapid, training-free evaluation protocol provides instant feedback. This synergistic framework transforms the search from a biased, unconstrained generation task into a structured, diverse, and efficient exploration. With our method, we obtain a more complete and dominant Pareto front of hardware-optimized models, achieving near-perfect quality scores. This is accomplished while dramatically reducing the search cost from multiple GPU-days, typical for supernet-based approaches, to mere minutes. The contributions are summarized as follows:

- To counteract LLM's inherent exploration bias, we propose a **Complexity-Driven Partitioning Engine**. This engine systematically decomposes the entire search space into disjoint subspaces, based on a tangible architectural complexity metric (e.g., the count of specific operators), ensuring a diverse, comprehensive exploration.

- Within each partitioned niche, our framework employs an **LLM-Powered Co-evolutionary Operator** to generate novel candidate architectures. This operator tasks an LLM with two synergistic functions. As illustrated in Figure 2, it reflects on the results from previous generations to continually update and refine a Co-evolve Knowledge Base of design heuristics. Then guided by this evolving knowledge base and the current Pareto-optimal parents, it performs intelligent mutation and crossover. This approach transforms the LLM from a simple generator into a stateful agent that learns and applies design principles, accelerating the discovery of superior solutions.

- Compared to conventional and unconstrained LLM-driven methods, our training-free framework discovers a more complete and dominant set of optimal trade-offs. This superiority is validated by two standard metrics: a significantly higher **Hypervolume (HV)**, indicating our solutions achieve broader coverage of the performance space with both superior and more diverse models, and a lower **Inverted Generational Distance (IGD)**, showing our discovered architectures are closer to the true optimal front. The experiments

demonstrate that PEL-NAS enables this with a search cost of minutes, in stark contrast to the days of GPU training required by supernet-based approaches.

## 2 RELATED WORK

**Hardware-Aware Neural Architecture Search (HW-NAS).** HW-NAS is fundamentally a Multi-Objective Optimization Problem (MOP), tasked with discovering a set of Pareto-optimal architectures that balance conflicting objectives like accuracy and latency Njor et al. (2025); Benmeziane et al. (2021a). Benchmarks such as HW-NAS-Bench Li et al. (2021) are instrumental in standardizing research by providing pre-computed, real-world hardware metrics, thus accelerating the development cycle. The field has been largely dominated by supernet-based (one-shot) methods Cai et al. (2019); Chu et al. (2021); Sakuma et al. (2023). The core idea is to amortize training costs by pre-training a single, large network that contains all sub-architectures. Works like FairNAS Chu et al. (2021) represent cornerstones of this paradigm. However, their primary drawback is the immense computational cost and the inherent cost-fidelity trade-off. Efforts to improve the ranking consistency of subnets, such as the strict fairness sampling in FairNAS Chu et al. (2021), often consolidate or even increase the high computational overhead (e.g., 10 GPU-days for one supernet). This fundamental dilemma motivates our exploration of training-free approaches.

**Training-Free NAS and Zero-Cost Proxies.** To mitigate high training costs, training-free NAS employs zero-cost (ZC) proxies to predict model performance from initialized networks Li et al. (2024). The proxy landscape is diverse, including gradient-based metrics like snip and synflow Lee et al. (2018); Tanaka et al. (2020), higher-order information such as Jacobcov and grasp Mellor et al. (2021), and topology-based scores like SED Wu et al. (2024); Lee & Ham (2024). However, the landmark NAS-Bench-Suite-Zero study Krishnakumar et al. (2022) shows that individual proxies can be fragile. This leads to a trend of ensembling them to leverage their complementary information for more robust rankings He et al. (2024); Cortês et al. (2025).

**LLM-Driven Architecture Search.** While LLMs are now used as powerful evolutionary operators in NAS Zheng et al. (2023); Nasir et al. (2024), current methods face two critical limitations. First, their reliance on benchmark-specific oracles for feedback on accuracy and latency hinders real-world applicability. The second, more fundamental issue is LLM's inherent exploration bias, which is analogous to mode collapse in generative models Kossale et al. (2022). This bias, often amplified by alignment tuning Zhang et al. (2025), results in low-diversity outputs that trap the search in narrow regions of the solution space.

**Evolutionary Algorithms and Niching for Diversity.** Evolutionary Algorithms (EAs), particularly Multi-Objective EAs like NSGA-II Deb et al. (2002); Lu et al. (2020), are a natural fit for HW-NAS due to their effectiveness in handling discrete, multi-objective search spaces Booysen & Bosman (2024); White et al. (2021). A central challenge in evolutionary computation is preventing premature convergence by maintaining population diversity Shir (2012). Niching is a classic technique developed for this purpose. It works by forming and maintaining multiple subpopulations (niches) in parallel, allowing the algorithm to explore different optimal regions simultaneously Shir (2012).

## 3 METHODOLOGY

Our method, PEL-NAS, overcomes the critical exploration bias of LLMs in HW-NAS while preserving the efficiency of training-free methods. As illustrated in Figure 2, our approach integrates three key components: a search space partitioning strategy to ensure diversity, an LLM-powered evolutionary engine for intelligent exploration, and a training-free evaluator to provide rapid feedback on accuracy and latency.

### 3.1 COMPLEXITY-DRIVEN SEARCH SPACE PARTITIONING

The primary obstacle to effective LLM-driven NAS is the model's inherent *exploration bias*, or *mode collapse*. This tendency is severely exacerbated when the LLM confronts the vast and unstructured design space of neural architectures. Faced with countless possibilities, an unconstrained LLM defaults to restricted, familiar designs, failing to discover the diverse range of trade-offs required for a complete Pareto front.

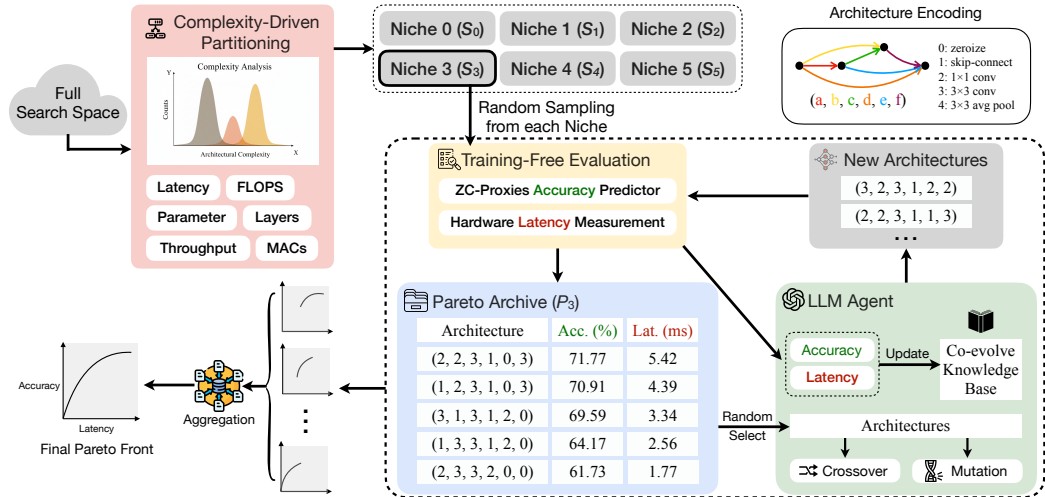

Figure 2: **The PEL-NAS framework.** The search space is partitioned into complexity-based niches, where an LLM performs parallel evolutionary search. The individual results are then aggregated to form the final, complete Pareto front, mitigating exploration bias.

To counteract this fundamental bias, we introduce Complexity-Driven Search Space Partitioning. Rather than searching the entire space, we divide the entire space into multiple, disjoint subspaces, or *niches*.

Our key insight is that this partitioning should not be arbitrary but must be rooted in a tangible architectural property that directly governs hardware performance. Our empirical analysis of the HW-NAS-Bench space (Figure 3) confirmed this, revealing a strong correlation between model complexity and the count of the most parameter-heavy operator: `nor_conv_3x3`. Intuitively, a $3\times3$ convolution introduces 9-times more kernel parameters per channel pair than a $1\times1$ convolution, so increasing the number of `nor_conv_3x3` blocks causes a step-like growth in parameters and typically in latency.

This finding provides a clear, data-driven rationale for our strategy. By partitioning the search space based on the count of `nor_conv_3x3` operators (Table 1), we create niches that correspond to meaningful families of architectural complexity. This forces the LLM to maintain dis-

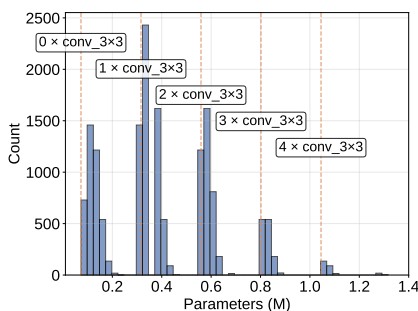

Figure 3: Analysis of the HW-NAS-Bench search space. The distribution of total parameters exhibits clear clustering, where each cluster corresponds to a specific number of `nor_conv_3x3`.

tinct populations across the entire complexity spectrum, from ultra-lightweight to highly complex, directly mitigating mode collapse and ensuring a comprehensive exploration.

Table 1: Complexity-driven partitioning of the search space into six disjoint niches. The partitioning strategy is designed to force exploration across the entire architectural complexity spectrum, from simple non-convolutional models to highly complex ones

| Niche | # 3x3 conv | # 1x1 conv | Rationale |
|---|---|---|---|
| Niche 0 ($\mathcal{S}_0$) | 0 | 0 | Explores non-convolutional architectures |
| Niche 1 ($\mathcal{S}_1$) | 0 | $\geq 1$ | Focuses on simple, low-latency models |
| Niche 2 ($\mathcal{S}_2$) | 1 | Any | Entry-level complex architectures |
| Niche 3 ($\mathcal{S}_3$) | 2 | Any | Mid-level complexity |
| Niche 4 ($\mathcal{S}_4$) | 3 | Any | High-level complexity |
| Niche 5 ($\mathcal{S}_5$) | $\geq 4$ | Any | Explores the most complex designs |

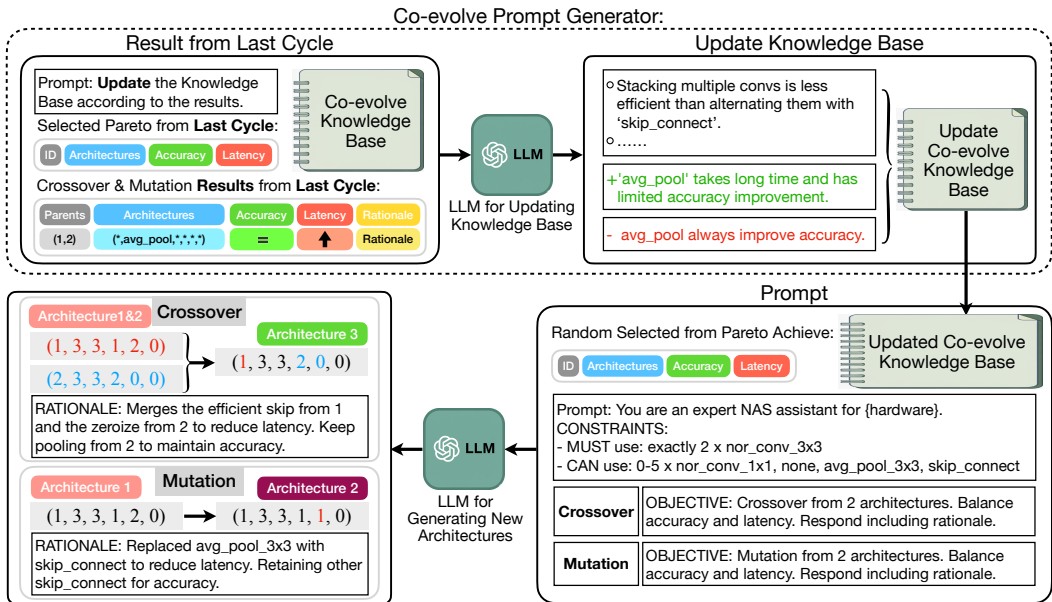

Figure 4: **The Co-evolve Prompt Generator in PEL-NAS.** The LLM first acts as a reasoning engine, updating a Knowledge Base by analyzing prior results. This learned knowledge then informs the LLM's second role as an expert architect, where it generates new, rationale-driven architectures under specific constraints, creating a self-optimizing search process.

## 3.2 LLM-POWERED PARTITIONED CO-EVOLUTION OF PROMPTS AND ARCHITECTURES

As illustrated in Figure 4, the **Co-evolve Prompt Generator** operates in two tightly coupled phases that realize the co-evolution of prompts and architectures.

**Knowledge Base Update** After each search cycle, PEL-NAS collects the architectures along with their measured *accuracy*, *latency* and the corresponding design rationales from previous cycle. The LLM first acts as a reasoning engine, analyzing these results to update a *Co-evolve Knowledge Base*. For example, the Knowledge Base may update rules such as "avg_pool takes a long time and has limited accuracy improvement" and delete "avg_pool always improves accuracy". By continuously summarizing such patterns, the LLM accumulates long-term memory of effective design principles and avoids repeatedly exploring unpromising regions, preventing local mode collapse.

**Rationale-driven Generation** The updated knowledge base is then injected into the next prompt, together with Pareto architectures selected from the archive, to guide the LLM's second role as an expert architect. Within this role, the LLM generates new candidate architectures through two operators: **1)Crossover:** merges components of two parent architectures to balance accuracy and latency. For instance, Figure 4 shows combining the skip connection from one parent with the zerorized block from another to reduce latency while preserving pooling layers for accuracy. **2)Mutation:** modifies a single architecture to further refine efficiency. For example, replacing avg_pool_3x3 with skip_connect lowers latency while retaining other beneficial connections.

## 3.3 TRAINING-FREE OBJECTIVE EVALUATION

An effective evolutionary search is critically dependent on rapid and reliable fitness feedback. Traditional model training is infeasible due to its prohibitive time cost, a bottleneck that has plagued recent LLM-driven methods like LLMatic Nasir et al. (2024), whose search costs can exceed even those of pre-trained supernet paradigms.

To avoid this, our framework relies on an efficient, training-free evaluation protocol. For each candidate architecture $A$, we assess two objectives: its hardware latency $l(A)$ and its predicted performance $z_{pred}(A)$. We obtain latency directly from the HW-NAS-Bench lookup table, which simulates rapid, noise-free hardware measurements. To estimate performance without costly training, we

employ a surrogate model, following the state-of-the-art ensemble strategy from Krishnakumar et al. (2022). Specifically, we use an XGBoost model that takes the full set of 13 zero-cost (ZC) proxies from NAS-Bench-Suite-Zero as input features. This predictor achieves a strong Spearman's rank correlation of approximately 0.90 with the ground truth, providing a reliable and efficient signal to guide the evolutionary search.

# 4 EXPERIMENTS

## 4.1 EXPERIMENTAL SETUP

**Datasets.** We use **HW-NAS-Bench** Li et al. (2021), a comprehensive benchmark that provides ground-truth accuracy on CIFAR-10, CIFAR-100, and ImageNet16-120 and latency measurements for 15,625 architectures across six real-world hardware devices: **Edge GPU (NVIDIA Jetson TX2)**, **Raspberry Pi 4**, **Edge TPU (Google TPU Dev Board)**, **Pixel 3**, **ASIC (Eyeriss)**, and **FPGA**. For the Vision Transformer (ViT) part of our study, we evaluate our framework on **ImageNet-1k**.

**Baselines.** We position PEL-NAS against a diverse set of state-of-the-art NAS methods to highlight its unique advantages. Our comparison includes influential supernet-based methods that do not primarily focus on hardware constraints, such as the classic differentiable approach **DARTS** Liu et al. (2018) and the fairness-enforcing **FairNAS** Chu et al. (2021). To benchmark against a hardware-aware contemporary, we include **PRP-NAS** Benmeziane et al. (2023), which represents supernet methods that explicitly optimize for hardware efficiency. Furthermore, we contrast our approach with the latest advancements in LLM-driven search by **LLMatic** Nasir et al. (2024), that also utilize large language models for architecture generation but are not designed with hardware awareness as a primary objective. For ViT on ImageNet-1k, we report **ViT-B/16** Dosovitskiy et al. (2020), **DeiT-B** Touvron et al. (2021), and the NAS search method **AutoFormer** Chen et al. (2021).

**Evaluation Metrics.** We evaluate the quality of the set of discovered solutions, known as a **Pareto front** ($S$), against the true, theoretically perfect front ($P^*$). Conceptually, a Pareto front represents the collection of *best possible trade-offs*. In our context, for any model on the front, no other model exists that is simultaneously more accurate *and* faster (lower latency). A superior search algorithm is one that discovers a front that is both high-quality and comprehensive. Evaluating the quality of a Pareto front is a nuanced task, as it requires assessing two distinct properties simultaneously:

To provide a holistic and robust evaluation, we use HV and IGD, two widely adopted standard metrics in multi-objective optimization that synergistically address these requirements. HV assesses the overall quality and spread of the discovered solutions, measuring the overall coverage of the discovered front, while IGD measures the fidelity of the found front by quantifying how closely its solutions approximate the ideal true optimal front.

- **Hypervolume (HV):** This metric measures the overall *coverage* of the discovered front. It rewards fronts that contain a wide variety of solutions that are both highly accurate and fast. Formally, given a reference point $r$ that is dominated by all solutions in the front $S$, the HV is the volume of the region bounded by the front and the reference point:

$$\text{HV}(S, r) = \text{volume}\left(\bigcup_{s \in S} [s_1, r_1] \times [s_2, r_2] \times \cdots \times [s_m, r_m]\right)$$

  A larger HV is better, indicating a more complete and higher-quality front.

- **Inverted Generational Distance (IGD):** This metric measures the *closeness* or *fidelity* of our discovered front to the true, perfect front. It essentially answers the question: On average, how far away is each theoretically perfect solution from the nearest solution we actually found? It is defined as:

$$\text{IGD}(S, P^*) = \frac{1}{|P^*|} \sum_{p^* \in P^*} \min_{s \in S} d(p^*, s)$$

  where $d(\cdot, \cdot)$ is the Euclidean distance. A lower IGD is better, signifying a more accurate approximation of the true optimal front.

**Implementation Details and Hyperparameter Settings.** We use GPT-4.1 as our LLM engine. The evolutionary search runs for 10 generations. The crossover probability $p_c$ is set to 0.5. For our ZC ensemble predictor, we use an XGBoost model trained on the 13 proxies from NAS-Bench-Suite-Zero Krishnakumar et al. (2022).

## 4.2 MAIN RESULTS

Table 2: Comparison of selected top structures of HW-NAS-Bench on CIFAR-10. Acc.=Top-1 Accuracy, Lat.=Latency

| Architecture | Edge GPU | | Raspberry Pi 4 | | Pixel 3 | | FPGA | |
|---|---|---|---|---|---|---|---|---|
| | Acc. (%) | Lat. (ms) | Acc. (%) | Lat. (ms) | Acc. (%) | Lat. (ms) | Acc. (%) | Lat. (ms) |
| DARTS | $68.30 \pm 0.08$ | 5.36 | $68.30 \pm 0.08$ | 45.36 | $68.30 \pm 0.08$ | 11.4 | $68.30 \pm 0.08$ | 7.32 |
| FairNAS | $93.23 \pm 0.18$ | 4.68 | $92.51 \pm 0.90$ | 34.15 | $92.40 \pm 0.15$ | 8.65 | $92.90 \pm 0.23$ | 5.12 |
| PRP-NAS-BA | $\mathbf{94.37 \pm 0.02}$ | 4.35 | $93.68 \pm 0.05$ | 40.7 | $94.20 \pm 0.03$ | 5.60 | $94.37 \pm 0.01$ | 6.80 |
| PRP-NAS-BL | $92.34 \pm 0.05$ | 2.30 | $88.70 \pm 0.03$ | 7.60 | $89.57 \pm 0.07$ | 3.60 | $91.35 \pm 0.04$ | 3.60 |
| LLMatic | $94.26 \pm 0.13$ | 6.80 | $94.26 \pm 0.13$ | 69.06 | $94.26 \pm 0.13$ | 21.59 | $94.26 \pm 0.13$ | 6.67 |
| **PEL-NAS (Ours)** | $\mathbf{94.37 \pm 0.02}$ | 4.35 | $\mathbf{94.37 \pm 0.15}$ | 69.76 | $\mathbf{94.30 \pm 0.15}$ | 21.59 | $\mathbf{94.37 \pm 0.14}$ | 6.68 |
| | $93.88 \pm 0.10$ | 3.36 | $92.37 \pm 0.07$ | 18.67 | $93.31 \pm 0.05$ | 8.98 | $93.29 \pm 0.15$ | 2.91 |
| | $89.18 \pm 0.15$ | **1.78** | $90.70 \pm 0.12$ | **7.29** | $90.36 \pm 0.08$ | **2.57** | $89.57 \pm 0.25$ | **1.65** |

Our main experimental results are detailed in Tables 2, 3 and 4. Collectively, they demonstrate that PEL-NAS not only discovers architectures that achieve a balance between accuracy and latency, a Pareto front of superior quality and completeness, but also achieves this with unparalleled efficiency.

**Analysis of Discovered Architectures of HW-NAS-Bench on CIFAR-10.** Beyond the overall front quality, the individual architectures in Table 2 highlight the practical value of our method. PEL-NAS not only finds models with state-of-the-art accuracy, matching the performance of costly supernet-based methods, but also excels in the low-latency domain where other approaches falter. Crucially, it discovers the undisputed fastest architecture for each hardware target. For example, it identifies a model with a latency of just 1.78ms on the Edge GPU and 1.65ms on the FPGA—outperforming the fastest competitor, PRP-NAS-BL, by over 22% and 54% respectively. This proves its superior ability to explore the full spectrum of trade-offs and deliver a truly comprehensive set of optimal solutions.

Table 3: HV and IGD comparison on HW-NAS-Bench across six hardware devices on CIFAR-10, CIFAR-100, and ImageNet16-120. PEL-NAS consistently outperforms all baselines, demonstrating its ability to find a more complete and dominant Pareto front. (Higher HV is better, lower IGD is better). Best results are in **bold**

| Method | Edge GPU | | Raspi 4 | | Edge TPU | | Pixel 3 | | Eyeriss | | FPGA | |
|---|---|---|---|---|---|---|---|---|---|---|---|---|
| | HV ↑ | IGD ↓ | HV ↑ | IGD ↓ | HV ↑ | IGD ↓ | HV ↑ | IGD ↓ | HV ↑ | IGD ↓ | HV ↑ | IGD ↓ |
| | | | | | **CIFAR-10** | | | | | | | |
| LLMatic | 0.191 | 0.542 | 0.549 | 0.296 | 0.354 | 0.514 | 0.551 | 0.337 | 0.512 | 0.331 | 0.586 | 0.370 |
| FairNAS | 0.892 | 0.073 | 0.962 | 0.035 | 0.947 | 0.089 | 0.971 | 0.033 | 0.958 | 0.068 | 0.918 | 0.091 |
| PRP-NAS | 0.843 | 0.116 | 0.926 | 0.133 | 0.916 | 0.123 | 0.926 | 0.124 | 0.928 | 0.145 | 0.903 | 0.241 |
| **PEL-NAS** | **0.997** | **0.006** | **0.997** | **0.013** | **0.955** | **0.057** | **0.996** | **0.011** | **0.961** | **0.037** | **0.931** | **0.046** |
| | | | | | **CIFAR-100** | | | | | | | |
| LLMatic | 0.233 | 0.571 | 0.516 | 0.411 | 0.455 | 0.465 | 0.745 | 0.256 | 0.552 | 0.297 | 0.598 | 0.241 |
| FairNAS | 0.853 | 0.072 | 0.930 | 0.058 | 0.929 | 0.102 | 0.930 | 0.055 | 0.952 | 0.110 | 0.958 | 0.117 |
| PRP-NAS | 0.794 | 0.161 | 0.824 | 0.179 | 0.751 | 0.190 | 0.817 | 0.174 | 0.863 | 0.246 | 0.798 | 0.317 |
| **PEL-NAS** | **0.992** | **0.009** | **0.994** | **0.016** | **0.981** | **0.017** | **0.985** | **0.023** | **0.962** | **0.050** | **0.977** | **0.032** |
| | | | | | **ImageNet16-120** | | | | | | | |
| LLMatic | 0.285 | 0.566 | 0.340 | 0.461 | 0.279 | 0.632 | 0.783 | 0.193 | 0.392 | 0.428 | 0.678 | 0.230 |
| FairNAS | 0.838 | 0.115 | 0.894 | 0.048 | 0.851 | 0.122 | 0.907 | 0.067 | 0.912 | 0.086 | 0.916 | 0.079 |
| PRP-NAS | 0.833 | 0.096 | 0.857 | 0.082 | 0.887 | 0.116 | 0.892 | 0.073 | 0.879 | 0.096 | 0.876 | 0.113 |
| **PEL-NAS** | **0.953** | **0.043** | **0.988** | **0.011** | **0.943** | **0.033** | **0.983** | **0.042** | **0.945** | **0.050** | **0.972** | **0.028** |

**Pareto Front Quality Evaluation with HV and IGD.** The core quantitative results in Table 3 compare the discovered Pareto fronts using HV and IGD. Across all three datasets and six hardware targets, PEL-NAS consistently and significantly outperforms all baselines. PEL-NAS achieves higher HV scores and the lower IGD scores compared with baselines. For example, on CIFAR-10, PEL-NAS can achieve up to 80.6% higher HV and 53.6% lower IGD compared with non-constrained LLM Method. On CIFAR-100, PEL-NAS outperforms LLMatic, FairNAS, PRP-NAS, by 46.5%, 5.7%, 17.4% in HV, and by 34.9%, 6.1%, 18.6% in IGD, in average respectively. These observations further confirm that the front discovered by PEL-NAS is not only larger in volume but also much closer to the true optimal front. The experimental results also demonstrate that our complexity-driven partitioning strategy is highly effective in mitigating the LLM's generative bias and enabling a more complete and diverse exploration of the search space.

**Search Cost.** Crucially, as shown in Table 4, PEL-NAS achieves these results with negligible computational cost. As a training-free method, its search cost is measured in API calls (120 times) and minutes, starkly contrasting with supernet-based methods like FairNAS Chu et al. (2021) that require days of GPU training. In contrast, LLMatic Nasir et al. (2024) is the most time-consuming because it needs to train every generated architecture from scratch. This

Table 4: Search Cost per Dataset per Device on a V100 GPU

| Architecture | Search Cost |
|---|---|
| LLMatic | 17 GPU Days |
| FairNAS | 10 GPU Days |
| DARTS | 4 GPU Days |
| PRP-NAS-BA | 2 GPU Days |
| **PEL-NAS (Ours)** | **3 mins (API Calls)** |

combination of superior search capability and extreme efficiency makes PEL-NAS a practical and powerful solution for real-world HW-NAS challenges.

## 4.3 ABLATION STUDIES

Table 5: Ablation study results on CIFAR-100 showing the impact of each component of PEL-NAS. Both the partitioning strategy and the ZC ensemble are shown to be critical components, with their removal causing the most significant performance degradation

| Method | Average HV ↑ | Average IGD ↓ |
|---|---|---|
| **PEL-NAS (Full Model)** | **0.978 ± 0.017** | **0.0246 ± 0.0132** |
| *Ablation Studies:* | | |
| - without Partitioning | 0.516 ± 0.155 | 0.3734 ± 0.1197 |
| - without LLM Operator (uses PEA) | 0.843 ± 0.075 | 0.1649 ± 0.0311 |
| - without ZC Ensemble (uses Synflow) | 0.819 ± 0.112 | 0.1717 ± 0.0381 |

To isolate the contribution of each key component of our framework, we conduct a series of ablation studies. The aggregated results are summarized in Table 5, while detailed line graphs illustrating the search process for three datasets across six devices are available in the Appendix (Figures 6, 7, and 8). The analysis reveals that the partitioning strategy is the most critical element. Removing it (– without Partitioning) leads to a catastrophic performance collapse, which provides direct evidence that our niching approach is essential for mitigating the LLM's mode collapse. Similarly, the ZC ensemble predictor is vital; replacing it with a single Synflow proxy (– without ZC Ensemble) causes a significant performance degradation, confirming that a robust performance signal is crucial to guide the search effectively. Finally, while the partitioned evolutionary algorithm (PEA) (– without LLM Operator) still performs well, it is clearly surpassed by the full PEL-NAS model. This demonstrates that the LLM acts as an intelligent operator, leveraging context to generate superior candidates and further enhancing search efficiency.

## 4.4 GENERALIZABILITY ON VISION TRANSFORMER SEARCH SPACES

To validate the generalizability of PEL-NAS beyond CNNs, we extend our framework to a Vision Transformer (ViT) search space derived from **AutoFormer** Chen et al. (2021). We conduct our hardware-aware search experiments on the **ImageNet** dataset. To ensure an efficient search process, we employ an accuracy predictor. Specifically, we adopt the Auto-Proxy predictor from ViT-Bench-

101 Wei et al. (2024), which achieves a strong Spearman's rank correlation of $91.01 \pm 2.63$ on this task, confirming its reliability for performance estimation. All reported accuracies and the resulting Pareto front in Figure 5 are based on the outputs of this predictor.

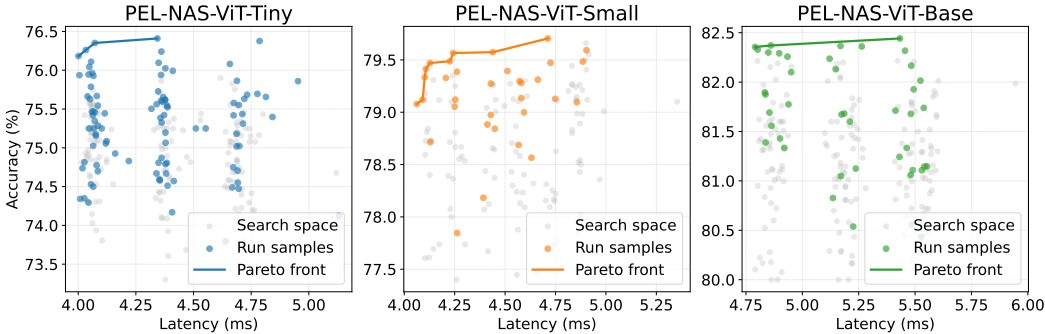

Figure 5: The Pareto front discovered by PEL-NAS for three AutoFormer search spaces on ImageNet. Latency is evaluated using a single NVIDIA A6000 GPU, and accuracy is estimated via a predictor

To create a realistic hardware-aware scenario, we profile the latency of each candidate architecture directly on our target device, a single NVIDIA A6000 GPU. We then apply the core principle of PEL-NAS—complexity-driven partitioning. Our analysis of the ViT architecture (see Appendix D for a detailed breakdown) reveals that computational complexity, a strong proxy for latency, is dominated by two key parameters: **Embed Dim** (quadratic impact, $O(D^2)$) and **Depth Num** (linear impact, $O(L)$). These parameters govern the scale of the MLP and the number of blocks, respectively, making them the most influential factors. We therefore partition the search space into niches based on discrete ranges of Embedding Dimension and Depth Number, enabling the LLM to efficiently explore trade-offs within structurally similar architectural families. The results, depicted in Figure 5 and detailed in Table 6, underscore the efficacy of our approach. PEL-NAS successfully identifies a dominant Pareto front, discovering architectures with superior accuracy-latency trade-offs.

Table 6: Comparison of Vision Transformer models found by PEL-NAS against state-of-the-art NAS methods. Latency is measured on A6000 GPU

| Method | Top-1 Acc (%) on ImageNet | Latency (ms) | Params (M) |
|---|---|---|---|
| ViT-B/16 Dosovitskiy et al. (2020) | 77.9 | 70 | 86 |
| DeiT-B Touvron et al. (2021) | 83.1 | 68 | 86 |
| AutoFormer Chen et al. (2021) | **83.4** | 8.4 | 23 |
| **PEL-NAS-ViT-Tiny (Ours)** | 76.2 | **4.0** | 6.9 |
| **PEL-NAS-ViT-Small (Ours)** | 79.7 | **4.7** | 16.1 |
| **PEL-NAS-ViT-Base (Ours)** | 82.5 | **5.4** | 20.2 |

## 5 CONCLUSION

In this work, we introduce PEL-NAS, a novel training-free framework designed to counteract the exploration bias inherent in LLM-driven neural architecture search. Our core contribution is a complexity-driven partitioning strategy that divides the search space into distinct niches, compelling the LLM to act as a parallel evolutionary engine and structurally enforcing population diversity across the entire architectural complexity spectrum. This approach effectively mitigates the LLM's tendency to converge on a narrow set of familiar architectures. Extensive experiments on HW-NAS-Bench demonstrate that PEL-NAS discovers a more complete and dominant Pareto front than baseline methods, validated by significantly superior HV and IGD scores. Our findings present a new paradigm for harnessing LLMs in combinatorial optimization, suggesting that imposing structural constraints on the generative process is a powerful method for mitigating inherent biases, future work could focus on automating the partitioning strategy and applying this framework to other complex design domains.

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

## A  LLM USAGE DISCLOSURE

We used large language models (LLMs) in two ways. (1) **Method component**: within PEL-NAS, an LLM serves as a co-evolutionary operator (Section 3.2) to generate candidates with rationale under niche constraints. (2) **Writing assistance**: we additionally used LLMs for minor editing (grammar, wording, and clarity). No generated text was used as scientific evidence without verification, and all experiments are fully reproducible from the described algorithms and released code.

## B  ALGORITHM

Algorithm 1 provides a detailed, step-by-step description of the PEL-NAS framework. The process begins with a one-time training of a zero-cost (ZC) ensemble predictor. The core of the algorithm is a parallel evolutionary search conducted independently within several disjoint niches ($\mathcal{S}_k$), which are defined by architectural complexity. In each generation, an LLM acts as an intelligent evolutionary operator to generate a new candidate architecture ($A_{child}$) under the niche-specific constraints. The candidate is then evaluated using the pre-trained predictor and direct hardware lookup, and the Pareto archive for that niche ($\mathcal{P}_k$) is updated. Finally, all niche archives are aggregated and filtered through a non-dominated sort to produce the final, comprehensive Pareto front.

---

**Algorithm 1** PEL-NAS: Partitioned Evolutionary LLM-driven NAS

---

1: **Input:** Number of generations $G$, LLM engine $\mathcal{L}$, niche definitions $\{\mathcal{S}_0, \ldots, \mathcal{S}_5\}$
2: **Output:** Final Pareto front $\mathcal{P}_{final}$

3: # Phase 1: Initialization
4: Train ZC ensemble predictor $\mathcal{M}_{pred}$ on a sample of architectures *// Offline, one-time step*
5: **for** $k \in \{0, 1, \ldots, 5\}$ **do**
6:     Initialize Pareto archive $\mathcal{P}_k \leftarrow \emptyset$
7:     Sample an initial population $Pop_{init} \subset \mathcal{S}_k$
8:     **for** each architecture $A \in Pop_{init}$ **do**
9:         $(z_{pred}, l) \leftarrow (\mathcal{M}_{pred}(A), \text{HardwareLookup}(A))$
10:         Update $\mathcal{P}_k$ with $(A, z_{pred}, l)$ *// Add if not dominated*
11:     **end for**
12: **end for**

13: # Phase 2: Partitioned Co-evolution
14: **for** generation $g = 1, \ldots, G$ **do**
15:     # Parallel evolution across all niches
16:     **for** $k \in \{0, 1, \ldots, 5\}$ **do**
17:         Select parent(s) $A_{parent}$ from $\mathcal{P}_k$
18:         Construct $Prompt$ using $A_{parent}$, their scores, and the constraint for niche $\mathcal{S}_k$
19:         Generate a new child architecture $A_{child} \leftarrow \mathcal{L}(Prompt)$
20:         **if** $A_{child}$ is valid, is novel, and satisfies constraint of $\mathcal{S}_k$ **then**
21:             $(z_{pred}, l) \leftarrow (\mathcal{M}_{pred}(A_{child}), \text{HardwareLookup}(A_{child}))$
22:             *// Update archive by adding the new solution and removing any it dominates*
23:             Let $A_{new} \leftarrow (A_{child}, z_{pred}, l)$
24:             $\mathcal{P}_k \leftarrow \{A' \in \mathcal{P}_k \mid A_{new} \text{ does not dominate } A'\} \cup \{A_{new}\}$
25:         **end if**
26:     **end for**
27: **end for**

28: # Phase 3: Final Aggregation
29: $\mathcal{P}_{union} \leftarrow \bigcup_{k=0}^{5} \mathcal{P}_k$
30: $\mathcal{P}_{final} \leftarrow \text{Non-Dominated-Sort}(\mathcal{P}_{union})$
31: **return** $\mathcal{P}_{final}$

---

# C   RESULT OF ABLATION STUDY ON ALL DATASETS AND DEVICES

This section provides a comprehensive visualization of the ablation studies discussed in the main paper's Section 4. We present the full set of Pareto fronts for each of the three datasets—CIFAR-10, CIFAR-100, and ImageNet16-120—across all six hardware devices from the HW-NAS-Bench benchmark. These figures visually supplement the aggregated quantitative results presented in Table 5 and demonstrate the consistent and crucial contribution of each component within the PEL-NAS framework.

In each subplot, the reader can clearly observe that the Pareto front discovered by the full PEL-NAS model (in blue) consistently envelops and dominates the fronts from the three ablated versions. This provides strong visual evidence that each key component of our framework—the partitioning strategy, the LLM operator, and the ZC ensemble predictor—is critical for discovering the optimal trade-off between accuracy and latency across diverse datasets and hardware constraints.

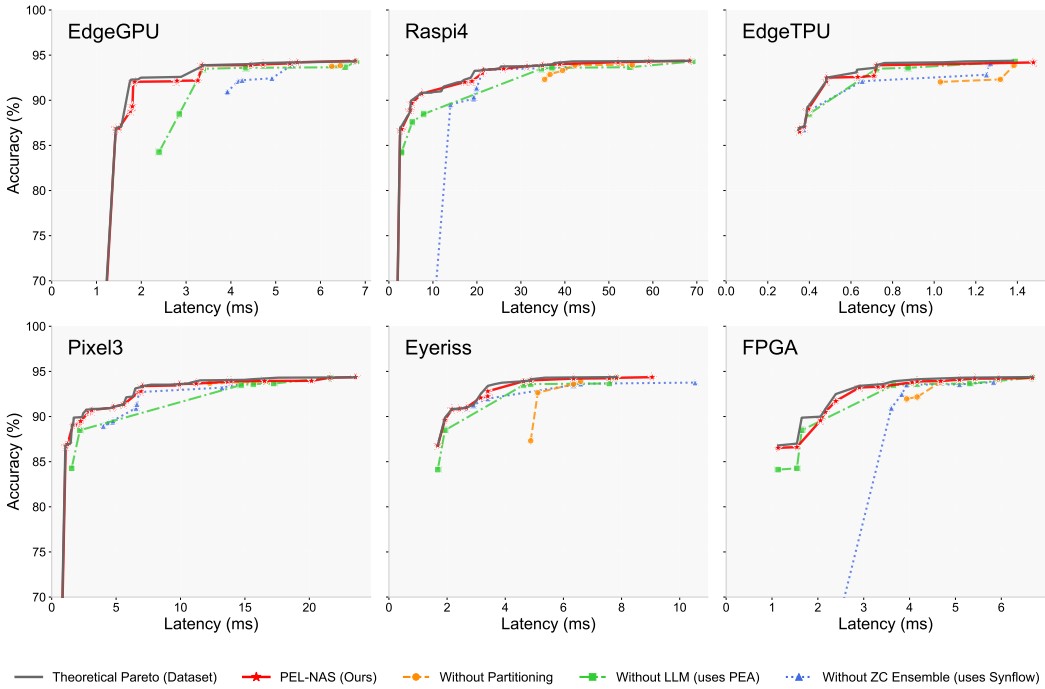

Figure 6: **Results of the ablation study on CIFAR-10 across six hardware devices.** Each subplot compares the Pareto fronts discovered by our full model (**PEL-NAS**) against its three ablated versions. The consistent dominance of the full PEL-NAS model demonstrates that each component is crucial for discovering the optimal trade-off between accuracy and latency.

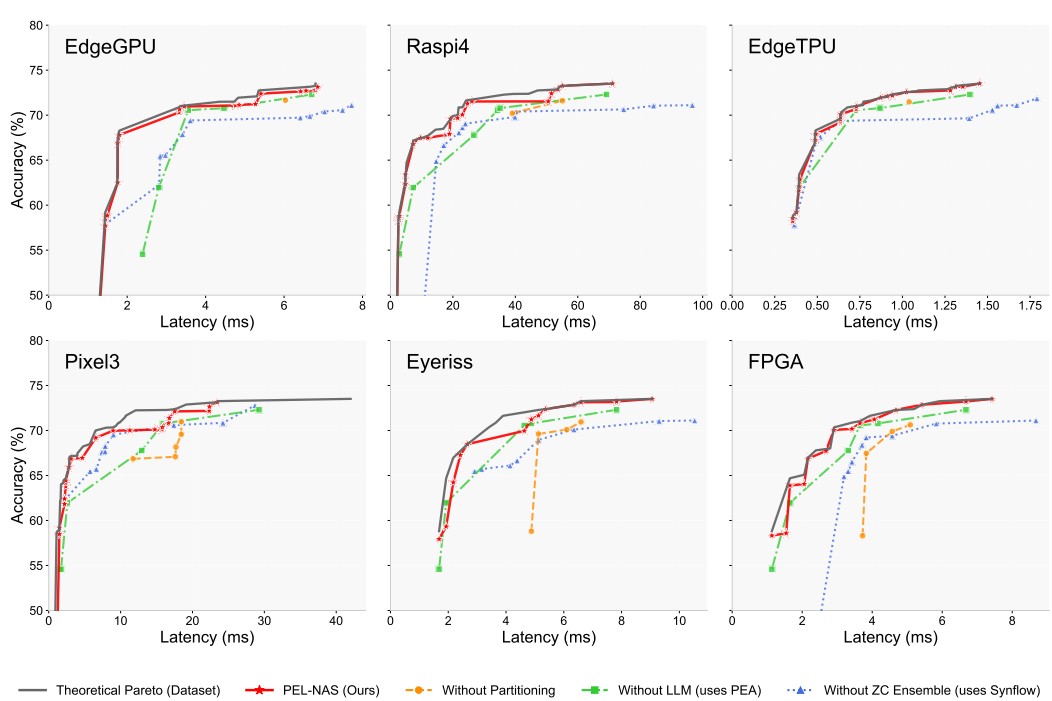

Figure 7: Results of the ablation study on CIFAR-100 across six hardware devices.

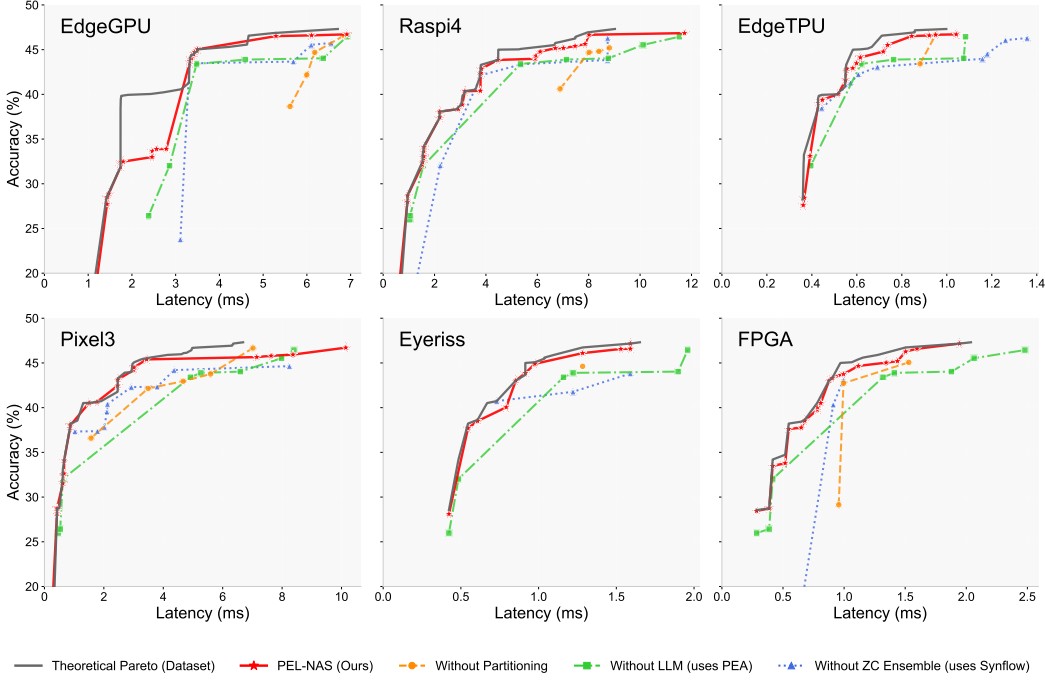

Figure 8: Results of the ablation study on ImageNet16-120 across six hardware devices.

# D COMPUTATIONAL COMPLEXITY ANALYSIS OF THE VISION TRANSFORMER SEARCH SPACE

To apply our complexity-driven partitioning strategy to the Vision Transformer (ViT) search space, we first conduct a formal analysis of how different architectural parameters influence the model's total computational load, measured in floating-point operations (FLOPs). This analysis provides a principled foundation for identifying the most impactful parameters, which are then used to define the disjoint niches for our search algorithm. The primary parameters in a ViT search space like AutoFormer's Chen et al. (2021) are **Embed Dim** ($D$), **Depth Num** ($L$), **MLP Ratio**, **Q-K-V Dim** ($D_h$), and **Head Num** ($h$).

A Transformer's computation is concentrated in two main components within each block: the Multi-Head Self-Attention (MHSA) module and the Multi-Layer Perceptron (MLP) module. A key feature of the AutoFormer search space is that it decouples the main Embed Dim ($D$) from the Q-K-V Dim ($D_h$) used within the attention mechanism.

The total FLOPs can be approximated by:

$$\text{Total FLOPs} \approx L \times (\text{FLOPs}_{\text{MHSA}} + \text{FLOPs}_{\text{MLP}})$$

ANALYSIS OF COMPONENTS

1. **Multi-Head Self-Attention (MHSA):** In the decoupled design, an input of size $N \times D$ (where $N$ is the number of patches) is projected to Q, K, and V tensors of size $N \times D_h$. The output is then projected back to $N \times D$.
    - **Q, K, V Projections:** $O(N \cdot D \cdot D_h)$
    - **Attention & Value Summation:** $O(N^2 \cdot D_h)$
    - **Output Projection:** $O(N \cdot D_h \cdot D)$

    The complexity of the MHSA block is thus jointly determined by $D$ and $D_h$.

2. **Multi-Layer Perceptron (MLP):** The MLP block operates on the main embedding dimension $D$. It typically consists of two linear layers, with the first expanding the dimension by the 'MLP Ratio' and the second projecting it back down.

    $$\text{FLOPs}_{\text{MLP}} \approx 2 \cdot N \cdot D \cdot (D \cdot \text{MLP Ratio}) = O(N \cdot D^2 \cdot \text{MLP Ratio})$$

PARAMETER IMPACT RANKING

Based on the combined formula, we can rank the parameters by their impact on computational complexity:

1. **Embed Dim** ($D$): This is the **most influential** parameter. Its impact is quadratic ($O(D^2)$) due to its role in the MLP block, which constitutes a significant portion of the total computation.

2. **Depth Num** ($L$): This parameter has a **direct linear impact** ($O(L)$) on the total FLOPs, as it multiplies the computation of the entire Transformer block. It is the second most influential factor.

3. **MLP Ratio:** This parameter has a **strong linear impact** by scaling the largest term in the complexity formula ($N \cdot D^2$).

4. **Q-K-V Dim** ($D_h$): In the decoupled architecture, this parameter has a **moderate linear impact** ($O(D_h)$), affecting only the MHSA module.

5. **Head Num** ($h$): This parameter has a **negligible impact** ($O(1)$) on FLOPs. For a fixed total 'Q-K-V Dim' ($D_h$), changing the number of heads only alters how the computation is parallelized, not the total amount.

This analysis provides a clear, principled rationale for our partitioning strategy. By creating niches based on **Embed Dim** and **Depth Num**, we are structuring the search around the two parameters that most fundamentally govern the model's computational complexity and, by extension, its hardware latency.

# E    ANALYSIS OF LLM EXPLORATION BIAS

This section provides the core visual evidence that motivates our partitioned search strategy. As demonstrated in Figure 9, when the LLM search is not structurally constrained by our partitioning scheme, its inherent exploration bias in generative models—becomes apparent. The LLM-generated architectures cluster heavily in a narrow region of the solution space, resulting in an incomplete and suboptimal Pareto front. This phenomenon powerfully illustrates that naive prompt engineering is insufficient to steer the LLM's generative process effectively, thereby underscoring the necessity of a structural intervention like our complexity-driven partitioning to achieve a comprehensive and diverse architecture search.

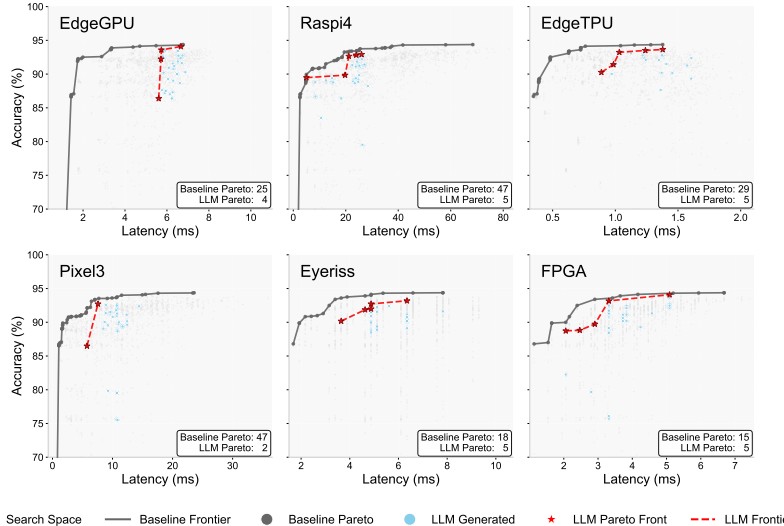

Figure 9: **LLM's mode collapse in NAS persists despite prompt engineering.** The figure shows the Pareto fronts discovered by an **unpartitioned** LLM-driven method, providing clear visual evidence of mode collapse. The LLM-generated architectures are **highly clustered in a narrow region of the performance-latency space**, resulting in a sparse and incomplete Pareto front that finds far fewer non-dominated solutions. This failure to explore—i.e., mode collapse—occurs even when the LLM is explicitly prompted to target diverse latencies, powerfully demonstrating the need for a more **structural intervention**, like our proposed partitioning strategy, to effectively guide the generative process.

# F  LLM PROMPT TEMPLATES AND CO-EVOLUTION PROCESS

This appendix provides the *full prompt structures* used in the two stages of each PEL-NAS generation and explains how they form a tight co-evolution loop.

STAGE 1: KNOWLEDGE-BASE UPDATE PROMPT

At the end of each generation, the LLM first acts as a **reasoning engine** to consolidate lessons learned from the previous search. It receives a prompt with the following explicit structure:

```
[System role]
You are a NAS analyst. Summarize design heuristics
for the given hardware-aware search space.

[Context]
- Target device and dataset: {device}, {dataset}
- Niche definition: {niche_constraints}
- Top Pareto parents from generation g:
  {list of parents with accuracy, latency, and rationales}

[Instruction]
1. Identify operator or connection patterns that
   consistently improve accuracy at acceptable latency.
2. Identify patterns that consistently hurt either metric.
3. Write explicit, concise rules of the form
   "Use/avoid ... because ...".
4. Remove or revise outdated rules that conflict with new evidence.

[Output format]
Return a JSON-like list called Updated_Knowledge_Base:
[
  {rule_1},
  {rule_2},
  ...
]
```

The output of Stage 1 is the updated *Co-evolve Knowledge Base* $\mathcal{K}_{g+1}$, which captures positive and negative architectural rules such as `"Prefer skip_connect after heavy conv layers to cut latency"` or `"Avoid multiple avg_pool_3x3 because they add latency with minimal accuracy gain"`.

STAGE 2: PROMPTED ARCHITECTURE GENERATION

Using $\mathcal{K}_{g+1}$, the LLM now plays the role of an **expert architect**. It receives a second, clearly structured prompt:

```
[System role]
You are an expert NAS designer that performs evolutionary
search inside a given niche under hardware constraints.

[Context]
- Target device and dataset: {device}, {dataset}
- Niche constraints: {niche_constraints}
  e.g., must contain exactly 2 × nor_conv_3x3,
        may contain any number of nor_conv_1x1,
        allowed ops: {allowed_ops}
- Current Pareto parents with metrics:
  {parent_1, parent_2, ...}
```

```
[Knowledge Base]
{Updated_Knowledge_Base from Stage 1}

[Evolution Operation]
Perform {N_child} new candidate generations.
For each child:
  * Decide Crossover or Mutation.
  * Describe exactly which blocks/edges you combine or modify.
  * Justify each change with expected effect on
    accuracy and latency (\le {latency_limit} ms).
  * Ensure all constraints are satisfied.

[Output format]
Return a list of JSON objects:
[
  {
    "child_id": "...",
    "operation": "crossover/mutation",
    "architecture_code": "...",
    "rationale": "..."
  },
  ...
]
```

**Niche-specific constraints.** The [Context] block above embeds the niche definition from Table 1. For example, the prompt for Niche 3 (exactly 2 nor_conv_3x3) includes:

```
Niche constraints:
- MUST use exactly 2 × nor_conv_3x3
- CAN use 0{4 × nor_conv_1x1
- ALLOWED operators: none, skip_connect, avg_pool_3x3
- Hardware latency must remain below {latency_limit} ms
```

Other niches simply change these numeric constraints while keeping the prompt skeleton identical.

**Integration of the two stages.** The LLM's Stage 2 output (new architectures and rationales) is immediately evaluated by the zero-cost predictor and hardware lookup. The resulting accuracy–latency pairs, together with rationales, are fed back into Stage 1 of the next generation:

$$\mathcal{K}_{g+1} \rightarrow \text{Stage 2 generation} \rightarrow \text{evaluation} \rightarrow \mathcal{K}_{g+2}.$$

This continuous feedback forms the **co-evolution of knowledge and prompts**, ensuring that each generation both (1) refines long-term design principles and (2) produces progressively better candidate architectures across all complexity-based niches.

