# OpenReview forum: "PEL-NAS: Search Space Partitioned Architecture Prompt Co-evolutionary LLM-driven Hardware-Aware Neural Architecture Search"
_ICLR.cc/2026/Conference — ICLR 2026 Conference Withdrawn Submission_

### Official Review · Reviewer_MXNW · 2025-10-22

**Soundness:** 2
**Presentation:** 3
**Contribution:** 2
**Rating:** 2
**Confidence:** 4

**Summary:**

In this paper, authors propose PEL-NAS, a training-free framework for Hardware-Aware Neural Architecture Search (HW-NAS) that addresses the exploration bias inherent in Large Language Model (LLM)-driven approaches. The authors observe that LLMs tend to generate architectures within limited regions of the search space, exhibiting a form of mode collapse. To address this issue, they propose a framework which contains three components: 1) a complexity-driven partitioning strategy that divides the search space into disjoint niches based on architectural complexity; 2) an LLM-based co-evolution mechanism that maintains and updates a knowledge base while generating new architectures; 3) a zero-cost ensemble predictor for rapid evaluation. Empirical results on HW-NAS-Bench demonstrate superior Pareto front with dramatically reduced search costs.

**Strengths:**

1. Authors identified a genuine problem in LLM-based NAS, i.e., exploration bias leading to incomplete Pareto fronts, as well as mode collapse. The complexity-driven partitioning strategy is empirically justified and provides an elegant solution to enforce diversity. Moreover, the complexity-driven partitioning is a kind of targeted structural intervention rather than just mere prompt engineering, demonstrates clear connection to hardware-related model complexity.

2. The dramatic reduction in search cost from GPU days to minutes while achieving promising results addresses a practical limitation in HW-NAS deployment. Besides, the successful extension to ViT search spaces demonstrates the framework’s adaptability beyond the original CNN-focused HW-NAS-Bench experiments.

3. The dual-stage prompt engineering that the LLM alternates between updating a knowledge base and generating architectures, represents a interesting and effective approach to leverage LLMs’ reasoning capabilities while maintain the search memory.

**Weaknesses:**

1. One of major concerns regarding this paper (as well as other similar LLM-driven NAS works) is the data contamination problem. For instance, HW-NAS-Bench contains only 15625 architectures and publicly available since 2021 (well before GPT-4’s training cutoff), while GPT-4 and other LLMs are trained on vast web corpora that likely include published NAS papers and their architectures. There is a substantial risk that the LLM might be essentially performing 'retrieval' rather than genuine 'search' or 'discovery'. Although the co-evolution mechanism and knowledge base updates might push the LLM slightly beyond memorisation, it would be better that authors can somehow verify the generated architectures are novel or outside the LLM’s training distribution.

2. Another concern is regarding the search space scalability, the manual identification of complexity-driving operators, e.g., nor_conv_3x3 for CNNs, Embed Dim and Depth Num for ViTs, raises questions about the scalability to novel search spaces. An automated/heuristic partitioning strategy would be more valuable.

3. While the authors identify exploration bias, they did not deeply analyse why LLMs exhibit this behaviour or explore prompt engineering alternatives that might directly address this bias without requiring partitioning. The incomplete analysis of LLM behaviour weakens their claimed corresponding contribution.

4. Some of compared baselines are outdated (e.g., DARTS is from 7 years ago), the proposed framework could benefit from comparisons with more recent training-free NAS methods, e.g., MeCo [1], SWAP [2] or L-SWAG [3], and other diversity-prompting techniques in evolutionary algorithms beyond the mentioned baselines.

&nbsp;
&nbsp;

[1] Jiang et al., MeCo: Zero-Shot NAS with One Data and Single Forward Pass via Minimum Eigenvalue of Correlation. NeurIPS 2023.

[2] Peng et al., SWAP-NAS: Sample-Wise Activation Patterns for Ultra-fast NAS. ICLR 2024.

[3] Casarin et al., L-SWAG: Layer-Sample Wise Activation with Gradients Information for Zero-Shot NAS on Vision Transformers. CVPR 2025.

**Questions:**

1. How sensitive is the proposed method to the number of niches? Have authors experimented with different partitioning granularities?

2. Can the partitioning strategy be automatically/heuristically learned rather than manually defined based on the search space analysis?

3. Can authors perform some memorisation tests? For example, prompt the LLM (e.g., GPT-4) to directly generate architectures from HW-NAS-Bench search space, and see whether it’s feasible, as well as the performance of produced architectures.

---

### Official Review · Reviewer_w62C · 2025-10-25

**Soundness:** 2
**Presentation:** 3
**Contribution:** 2
**Rating:** 4
**Confidence:** 4

**Summary:**

The paper proposes PEL-NAS, a hardware-aware NAS framework that (i) partitions the search space by simple architectural complexity (e.g., counts of 3×3 convs) to ensure diverse exploration, and (ii) uses an LLM with a persistent knowledge base to co-evolve design rules and prompts for generating candidates.

Candidates are scored without training: accuracy comes from an offline-trained surrogate built on zero-cost proxies, and latency from HW-NAS-Bench; multi-objective selection advances a Pareto set (HV/IGD). On CIFAR-10/100 and ImageNet-16-120 across six devices, PEL-NAS reports higher HV/lower IGD than prior baselines with minutes-level search; a ViT study adapts partitioning to depth/embedding size and uses Auto-Proxy plus measured latency.

Ablations suggest partitioning and the surrogate contribute the largest gains, with smaller incremental benefit from the LLM co-evolution.

**Strengths:**

- **Originality.** The paper identifies a clear and highly relevant problem: the inherent exploration bias (or mode collapse) of LLMs when applied to the vast NAS search space . The proposed solution, complexity-driven partitioning , is a novel and direct structural intervention to mitigate this specific, LLM-centric bias. This is combined with an LLM that maintains a persistent knowledge base to steer evolution, forming a targeted and well-motivated framework.
- **Quality.** Consistent Pareto gains (HV↑/IGD↓) across multiple devices and datasets; ablations clearly identify partitioning and the ZC-ensemble surrogate as principal contributors, with LLM co-evolution adding a smaller but positive increment.
- **Clarity.** The paper is well written and effectively uses clear figures (e.g., Figure 2 , Figure 4 ) and well-organized tables (e.g., Table 1 , Table 5 ) to present its methodology and results.

**Weaknesses:**

**Positioning and novelty.**
The paper's contributions center on two components, but the ablation study (Table 5) clearly shows that the search space partitioning is the paper's single most critical component; its removal causes the most significant performance degradation . In contrast, the LLM-KB (a form of persistent memory) has precedents in prior work (e.g., LLMatic’s [1] two archives include a prompt archive that stores/updates prompts over the search; RZ-NAS [2] adds explicit reflection modules), making its novelty more incremental.
While partitioning proves essential, a deeper analysis of its generality and robustness would strengthen the paper. Specifically:
- Lack of Analysis for Generality and Robustness:
  - The partitioning rules appear to be manually defined and may depend on domain-specific heuristics. For instance, the CNN search space is partitioned by the count of nor_conv_3x3 operators, whereas the ViT variant uses entirely different manually designed criteria, Embed Dim and Depth Num. The paper assumes that the number of 3×3 convolutions reliably reflects latency, but this “parameter ≠ latency” paradox is a well-known issue. This raises an open question about whether the current partitioning rule would remain effective if the optimization target were memory or another hardware metric. In such cases, should the partitioning instead be based on memory-bound operators?
  - While the proposed rule works well under the evaluated settings, its stability and generality across architectures remain uncertain. A targeted sensitivity study that varies both the partitioning axis (e.g., FLOPs, latency, parameter count, or memory usage) and the number of partitions, especially under different search-space scales, would help clarify how robust the framework truly is. In particular, comparing the proposed complexity-based partitioning against a random or uniform partition baseline would help confirm whether the performance gains stem from the partitioning principle itself or merely from enforcing any form of structured diversification. More broadly, developing an automated way to infer salient complexity dimensions for new search spaces would make this approach more generalizable and less dependent on manual expert analysis.

This suggests that the partitioning strategy is an orthogonal contribution, largely independent of the specific LLM-KB searcher. Demonstrating such generality, for example, by applying the partitioning method to other LLM-based searchers, would substantially strengthen the contribution and position it as a general tool for mitigating LLM exploration bias rather than just one component of a single method.

**Clarification of "Training-Free" Terminology.**
The "training-free" terminology warrants clarification. The method is only training-free during the search phase. Its accuracy estimation relies entirely on a pre-trained accuracy surrogate model  that must be trained offline. This is explicitly an XGBoost model fit on ZC proxies for CNNs  and a pre-existing "Auto-Proxy" predictor for ViTs. This approach is distinct from fully training-free methods that use raw ZC proxies directly for ranking without fitting a surrogate, and this distinction should be made clearer.

**Conflated Comparison of Search vs. Estimation Strategies.**
The main experimental comparisons (Tables 2 & 3) conflate the paper's search strategy (partitioned LLM) with the performance estimation strategy (surrogate model). PEL-NAS is benchmarked against methods using fundamentally different estimators, like supernets (FairNAS , PRP-NAS , and DARTS) or full-training (LLMatic). The resulting cost differences (Table 4) are largely dominated by the choice of estimator (surrogate vs. supernet), which reflects differences in methodological setup rather than isolating the contribution of the proposed search strategy.

Since the proposed LLM-based searcher with search-space partition could, in principle, operate on top of any performance estimator, whether a pre-trained supernet, a learned surrogate (as used in this paper), or even a raw zero-cost proxy score, contrasting it directly against supernet-based pipelines obscures what the LLM searcher itself contributes. A more precise evaluation would hold the performance estimator constant (e.g., using the same trained supernet or surrogate predictor) and compare searchers head-to-head, including Random Search, standard Evolutionary Algorithms, and prior LLM-based approaches such as LLMatic. This would more clearly isolate the effectiveness and efficiency of the proposed partitioned LLM search strategy itself.

[1] Muhammad U. Nasir, Sam Earle, Christopher Cleghorn, Steven James, Julian Togelius. LLMatic: Neural Architecture Search Via Large Language Models And Quality Diversity Optimization. GECCO 2024.

[2] Zipeng_Ji, Guanghui Zhu, Chunfeng Yuan, Yihua Huang. RZ-NAS: Enhancing LLM-guided Neural Architecture Search via Reflective Zero-Cost Strategy. ICML 2025.

**Questions:**

**On Sensitivity to the Partitioning Axis and Granularity.**
- The paper's CNN partitioning strategy is based on the nor_conv_3x3 count , which is identified as the most parameter-heavy operator. Is the general principle simply to partition by the most computationally expensive operator? How would the results change if a different operator, such as nor_conv_1x1, were used as the partitioning axis instead?
- The paper uses a fixed number of six niches for the HW-NAS-Bench space. How sensitive is the method's final performance (e.g., HV and IGD) to this choice? What would be the impact of using significantly fewer (e.g., 3) or more (e.g., 10) niches? Furthermore, should the optimal number of niches be relative to the overall size and complexity of the search space?
- As a diagnostic baseline, how does the proposed complexity-based partitioning compare against a random or uniform partition of the same size? Such a comparison could help determine whether the gains arise from the complexity metric itself or from structured diversification in general.

**On Fair Comparisons and the Generality of Partitioning.** To properly isolate the value of the proposed search strategy from the surrogate-based estimator:
- Could the authors provide results for baselines, including simple ones (e.g., Random Search, standard Evolutionary Algorithm) and, especially, searchers from prior LLM works (like LLMatic or RZ-NAS), that all use the exact same pre-trained ZC-proxy surrogate  for evaluation?
- More importantly, since the partitioning strategy appears to be an orthogonal contribution , could the authors apply this partitioning scheme to other prior LLM searchers (like LLMatic's) to demonstrate that it provides a general and consistent boost to their performance?

---

### Official Review · Reviewer_xvWB · 2025-10-31

**Soundness:** 3
**Presentation:** 3
**Contribution:** 2
**Rating:** 4
**Confidence:** 4

**Summary:**

This paper proposes PEL-NAS, a search-space partitioned, architecture prompt co-Evolutionary and LLM-driven NAS. The approach relies on complexity aware partitioning of the search space, an LLM-powered Evolutionary operating guided by an evolving knowledge base, and a training-free evaluation protocol. Experiments on HW-NAS-Bench shows good coverage of the Pareto front and good computational search performance compared to baselines.

**Strengths:**

1. PEL-NAS is a sensible NAS approach that demonstrates strong empirical performance and efficiency.

**Weaknesses:**

1. The partitioning method is manual, i.e., heavily reliant on the architecture search space. For example, the authors choose conv 3x3 following careful analysis of HW-NAS-Bench which of course doesn't apply to ViTs, so choose Embed Dim and Depth Num for ViT. This is a critical limitation of the method.
2. Related to the previous point, partitioning is critical to PEL-NAS (Table 5) but it is not clear how sensitive the choice of partitioning (partitioning criteria) is on performance.
3. The training-free objective evaluation, which leads to the efficient search cost, is not novel and is tied to the choice of benchmark,

**Questions:**

1. What is the impact of using a different LLM or variants of the prompt on the performance?
2. What is an example actual knowledge base produced?

---

### Official Review · Reviewer_ygWA · 2025-11-03

**Soundness:** 3
**Presentation:** 2
**Contribution:** 2
**Rating:** 4
**Confidence:** 4

**Summary:**

This paper introduces a novel framework for HW-NAS by partitioning the search space into complexity-based niches and performs an LLM as an evolutionary operator (crossover + mutation) whose prompts and design heuristics co-evolve from round-to-round; candidates are scored training-free using zero-cost proxies to target accuracy–latency Pareto fronts. The experiments are performed on standard benchmarks, demonstrating the effectiveness of PEL-NAS.

**Strengths:**

- A novel framework for NAS which used LLM to guide the search process.
-  Strong results on HW-NAS-Bench and ViT search spaces.

**Weaknesses:**

- Lack of theoretical support for the LLM-based section.  How and why is an LLM-based approach superior to traditional evolutionary search methods? Compared to evolutionary search, LLM-based search is expensive since it requires a pretrained language model.
- The idea of partitioning the search space into multiple subspaces is not novel, as it has been proposed in prior studies [1]
- Lack of ablation studies using other LLM models such as DeepSeek or Gemini.
- Lack of providing the performance curve as the search progresses.
- The reviewer believes that the framework is general. Instead of focusing on multiple objectives (i.e. accuracy and latency), how about benchmarking this method using accuracy as the sole objective? The authors should compare its performance against Random Search, Reinforcement Learning, and Evolutionary Search, using both single-proxy and ensemble-proxy settings, with and without search space partitioning, to validate the effectiveness of the LLM-based algorithm.



[1] Few-Shot Neural Architecture Search, Yiyang Zhao et al.

**Questions:**

See weaknesses.

---

### Note · Authors · 2025-12-03

I have read and agree with the venue's withdrawal policy on behalf of myself and my co-authors.